# Enhanced Efficacy of Gastric Cancer Treatment through Targeted Exosome Delivery of 17-DMAG Anticancer Agent

**DOI:** 10.3390/ijms25168762

**Published:** 2024-08-12

**Authors:** Jung Hyun Park, Say-June Kim, Ok-Hee Kim, Dong Jin Kim

**Affiliations:** 1Department of Surgery, Eunpyeong St. Mary’s Hospital, College of Medicine, The Catholic University of Korea, Seoul 03312, Republic of Korea; angle49@catholic.ac.kr; 2Department of Surgery, Seoul St. Mary’s Hospital, College of Medicine, The Catholic University of Korea, Seoul 03312, Republic of Korea; sayjunekim@gmail.com (S.-J.K.); ok6201@hanmail.net (O.-H.K.); 3Catholic Central Laboratory of Surgery, College of Medicine, The Catholic University of Korea, Seoul 03312, Republic of Korea; 4Translational Research Team, Surginex Co., Ltd., Seoul 06591, Republic of Korea

**Keywords:** bioengineered exosomes, 17-DMAG, MKN45 protein, gastric cancer, targetability

## Abstract

In this study, we explored the potential of genetically engineered exosomes as vehicles for precise drug delivery in gastric cancer therapy. A novel antitumor strategy using biocompatible exosomes (Ex) was devised by genetically engineering adipose-derived stem cells to express an MKN45-binding peptide (DE532) on their surfaces. 17-(Dimethylaminoethylamino)-17-demethoxygeldanamycin (17-DMAG) was encapsulated in engineered exosomes, resulting in 17-DMAG-loaded DE532 exosomes. In both in vitro and in vivo experiments using mouse gastric cancer xenograft models, we demonstrated that 17-DMAG-loaded DE532 Ex exhibited superior targetability over DE532 Ex, 17-DMAG-loaded Ex, and Ex. Administration of the 17-DMAG-loaded DE532 Ex yielded remarkable antitumor effects, as evidenced by the smallest tumor size, lowest tumor growth rate, and lowest excised tumor weight. Further mechanistic examinations revealed that the 17-DMAG-loaded DE532 Ex induced the highest upregulation of the pro-apoptotic marker B-cell lymphoma-2-like protein 11 and the lowest downregulation of the anti-apoptotic marker B-cell lymphoma-extra large. Concurrently, the 17-DMAG-loaded DE532 Ex demonstrated the lowest suppression of antioxidant enzymes, such as superoxide dismutase 2 and catalase, within tumor tissues. These findings underscore the potential of 17-DMAG-loaded DE532 exosomes as a potent therapeutic strategy for gastric cancer, characterized by precise targetability and the potential to minimize adverse effects.

## 1. Introduction

Exosomes are natural nano-sized cell-secreted vesicles that vary in size from 30 nm to 150 nm. They can transport various biomolecules and play important roles in the interactions with the immune system, making them highly suitable for precise drug delivery [1]. In recent years, advancements in exosome engineering technology have generated substantial interest in using targeted exosomes as a promising avenue for delivering therapeutics focusing on cancer treatment [2,3,4]. These attributes include biocompatibility, low immunogenicity, and an intrinsic capacity for transporting biomolecules, including proteins, lipids, and nucleic acids, between cells [5,6,7]. Furthermore, exosomes can traverse biological barriers, thereby establishing their suitability for the targeted delivery of diverse therapeutic agents, such as small molecules, siRNAs, and miRNAs, to specific cellular targets within an organism [8,9].

17-Dimethylaminoethylamino-17-demethoxygeldanamycin (17-DMAG) is a potent heat shock protein 90 (HSP90) inhibitor used to reshape cancer treatment [10,11,12]. HSP90 is a chaperone protein that plays a pivotal role in stabilizing and properly folding various client proteins, many of which are integral to cell growth, proliferation, and survival pathways [11]. By inhibiting HSP90, 17-DMAG disrupts the G2/M cell proliferation cycle, indicating profound implications in cancer therapy [13,14]. Its ability to orchestrate cell cycle arrest, induce apoptosis, and effectively curtail tumor proliferation underscores its potential for enhancing anticancer efficacy. The antitumor effects of 17-DMAG on gastric cancer cell lines have previously been explored [10]. However, a targeted delivery system might help increase the efficacy of anticancer treatments and reduce their toxicity.

Gastric cancer is a global burden, being the fifth most common cancer and the third most common cause of cancer-related death globally [15]. Although many cytotoxic chemotherapy agents and targeted and immune-modulatory agents have shown strong performance in cancer treatment, drug resistance issues remain. Therefore, novel approaches to cancer treatment are essential. To obtain optimally targeted exosomes, we needed to identify a specific target expressed on the membrane of the gastric cancer cell line. MKN45 is one of the most common gastric cancer cell lines and expresses many biomarkers related to tumor progression [16,17,18]. Among the studies of the biomarkers within MKN45, a study by Sahin et al. [19] employed a phage display peptide library to screen for proteins with a specific binding affinity for MKN45 gastric cancer cells. Their research established that the DE532 peptide (VETSQYFRGTLS) exhibited specific binding capabilities in MKN45 gastric cancer cells.

In this study, we proposed that 17-DMAG-loaded MKN45-targeting exosomes effectively deliver 17-DMAG to gastric cancer cells, resulting in enhanced antitumor effects. This innovative approach leverages the advantages of exosome-based delivery systems, such as biocompatibility and the ability to cross biological barriers, while exploiting MKN45-specific targeting of tumor cells. Additionally, we conducted the same experiments using the adenocarcinoma gastric (AGS) cell line to explore the efficacy of 17-DMAG-loaded DE532 exosomes in a gastric cancer cell line other than MKN45. Our research aims to provide insights into the design of targeted exosome-based therapies for gastric cancer in terms of potential clinical applications.

## 2. Results

### 2.1. Generation and Characterization of 17-DMAG-Loaded DE532 Exosomes

Building upon the expression of the DE532 target in MKN45 cells, in the present study, a pDisplay vector containing the genetic sequence encoding the DE532 peptide was used for transfection into adipose-derived stem cells (ASCs) to engineer exosomes capable of expressing the DE532 peptide. Finally, 17-DMAG was encapsulated within the DE532-expressing exosomes using electroporation (Figure 1A).

First, we compared the morphological differences between exosomes obtained from non-transfected ASCs (Ex) and transfected ASCs (tEx) (Figure 1B). Transmission electron microscopy (TEM) images showed no apparent visual differences in the nanoparticle-like structures between the two groups. Furthermore, using nanoparticle tracking analysis (NTA) with ZetaView, Ex exhibited a mean diameter of 141.6 nm and a concentration of 4.5 × 10^10^/mL, whereas tEx displayed a mean diameter of 151.5 nm and a concentration of 3.4 × 10^8^/mL.

Next, we compared the expression of exosome marker CD81 and Myc (a marker for the presence of the DE532 peptide) in Ex and tEx using flow cytometry (Figure 1C). CD81 was expressed at 64.2% in Ex and 53.4% in tEx, whereas the Myc marker (indicative of DE523 peptide expression) was expressed at 8.1% in Ex and 63.6% in tEx. This confirmed that tEx possessed the characteristics of exosomes and carried DE532 peptides.

### 2.2. In Vitro Antitumor Effects of 17-DMAG-Loaded DE532 Exosomes

Ex and tEx loaded with 50 nM 17-DMAG via electroporation were designated as Ex(D) and tEx(D), respectively. To investigate the antitumor effects of tEx(D), two types of gastric cancer cell lines, MKN45 and AGS, were treated with varying concentrations of 17-DMAG encapsulated into the exosomes, and the viability of the cancer cells was assessed using an MTT assay. The results revealed that tExo(D) caused a concentration- and time-dependent decrease in the viability of the MKN45 cells (Figure 2A). This trend was more pronounced in the AGS cells (Figure 2B).

Furthermore, we investigated the changes in the apoptotic markers when each set of materials, Ex, Ex(D), tEx, and tEx(D), was administered in the respective cell lines, using western blotting. In each experimental group, 5 × 10^6^ exosome particles were administered. For the encapsulated groups, Ex(D) and tEx(D), the concentration of encapsulated 17-DMAG was 40 nm. In the MKN45 cells, treatment with tEx(D) resulted in the most significant increase in the expression of the pro-apoptotic marker B-cell lymphoma 2 associated X, apoptosis regulator (BAX), and the greatest decrease in the expression of the anti-apoptotic marker B-cell lymphoma-extra large (Bcl-xL) (Figure 2C) (*p* < 0.05). The same trend was observed in the AGS cells (Figure 2D).

### 2.3. Effects of 17-DMAG-Loaded DE532 Exosomes on Mitochondrial Reactive Oxygen Species (ROS) and Antioxidant Enzymes

Changes in the mitochondrial ROS were investigated in the gastric cancer cell lines treated with each type of exosome: Ex, Ex(D), tEx, and tEx(D). MitoSOX staining was performed, and the signal intensities were compared. Compared to the control group, the Ex(D) and tEx(D) groups exhibited an increase in the MitoSOX fluorescence intensity, with the tEx(D) group showing the most significant increase (*p* < 0.05) (Figure 3A). This trend was also observed in the AGS cells (Figure 3B).

Furthermore, the expression of antioxidant enzymes in each cell line after treatment was compared using Western blot analysis. In the MKN45 cells, the treatment with tEx(D) resulted in the lowest expressions of superoxide dismutase (SOD), catalase, and glutathione peroxidase among all groups (Figure 3C). Similarly, in the AGS cells, the tEx(D) group exhibited the lowest expression of the antioxidant enzymes (Figure 3D).

### 2.4. In Vivo Validation: Effects of 17-DMAG-Loaded DE532 Exosomes on Tumor Size and Weight

In vivo mouse cancer models were used to compare the therapeutic effects of the treatments. A mouse model of gastric cancer was established by subcutaneously injecting MKN45 cells (5 × 10^6^ cells) into the flanks of immunocompromised mice. Subsequently, via tail vein injections, 10 × 10^10^ particles of each material were administered twice a week for 3 weeks (a total of six times), and changes in tumor size were observed. Thirty days after injections, the mice were euthanized, and the tumors were excised to measure their size and weight. When comparing tumor sizes among groups, the tEx(D) group exhibited the smallest tumor size (Figure 4A). Additionally, the tEx(D) group consistently showed the lowest tumor weight and volume (*p* < 0.05). Similarly, the therapeutic effects following treatment were compared in a xenograft mouse model using AGS cells. The tEx(D) group demonstrated the most significant reductions in tumor size and volume (Figure 4B).

### 2.5. In Vivo Validation: Effects of 17-DMAG-Loaded DE532 Exosomes on Apoptosis and Expression of Antioxidant Enzymes

To examine apoptosis following each treatment, tumor tissues were extracted, and real-time polymerase chain reaction (PCR) was conducted. Using real-time PCR, we investigated the mRNA changes in the apoptosis-related markers within the tumor tissues obtained from each treatment group. In the MKN45 cell xenograft model, treatment with tEx(D) significantly upregulated the expression of the pro-apoptotic marker Bax and downregulated the expression of the anti-apoptotic marker Bcl-xL to the greatest extent (*p* < 0.05) (Figure 5A). Similar trends were observed in the AGS cells (*p* < 0.05) (Figure 5B). Subsequently, Western blot analysis was performed to assess the changes in the apoptosis-related proteins within the tumor tissues obtained from each treatment group. In the MKN45 cell xenograft model, the tEx(D) treatment significantly upregulated the expression of Bax and downregulated Bcl-xL compared to the other treatment groups (*p* < 0.05) (Figure 5C). This pattern was also observed in the AGS cells (*p* < 0.05) (Figure 5D).

The expression of apoptosis-related factors was compared in tumor tissues from each group using immunohistochemistry. In the MKN45 cell xenograft model, tEx(D) treatment resulted in the most significant increase in the expression of the pro-apoptotic marker B-cell lymphoma-2-like protein 11 (BIM) and the most substantial decrease in the expression of Bcl-xL (*p* < 0.05) (Figure 6A). This pattern was also observed in the AGS xenograft model (Figure 6B). The expression of antioxidant enzymes in each group was compared using immunohistochemical staining. In the tumor tissues obtained from the MKN45 cell xenograft model, the tEx(D) treatment led to the most substantial reduction in the expressions of the antioxidant enzymes SOD2 and catalase (*p* < 0.05) (Figure 7A). Similar trends were observed in the AGS cell xenograft model (*p* < 0.05) (Figure 7B).

## 3. Discussion

The current research involves the encapsulation of conventional therapeutic drugs within nanoparticles designed to target gastric cancer specifically. This approach is expected to enhance the therapeutic efficacy against gastric cancer and significantly reduce adverse effects. This study investigated the anticancer effects of genetically engineered exosomes loaded with 17-DMAG for treating gastric cancer. The targetability and efficacy of the intravenously administered 17-DMAG-loaded DE532 exosomes [10] were investigated in vitro and in vivo using mouse gastric cancer xenograft models. tEx(D) demonstrated higher targetability than that of other materials, such as tEx, Ex(D), and Ex, and showed promising antitumor effects, as evidenced by the smallest tumor size, lowest tumor growth rate, and lowest weight of excised tumors. Moreover, tEx(D) administration resulted in the highest upregulation of the pro-apoptotic marker BIM and the lowest downregulation of the anti-apoptotic marker Bcl-xL, with the lowest expression of antioxidant enzymes, such as SOD2 and catalase, within tumor tissues derived from mouse gastric cancer xenograft models. Consequently, 17-DMAG-loaded DE532 exosomes are believed to exhibit the highest anticancer efficacy by achieving targetability in gastric cancer.

The need for new anticancer treatments has arisen from the limitations of existing therapies. Cytotoxic chemotherapeutic agents, such as 5-fluorouracil, cisplatin, and oxaliplatin, are widely used to treat stomach cancer because of their effectiveness in attacking cancer cells. However, these treatments also affect normal cells, leading to various side effects [20]. Targeted treatments have been developed to alleviate some of these side effects; however, problems remain. For example, trastuzumab for human epidermal growth factor receptor 2-positive gastric cancer and ramucirumab for advanced gastric cancer have shown improved results compared to those of traditional chemotherapy. However, they are not applicable to all patients, are expensive, and have considerable resistance issues, similar to those of other cytotoxic anticancer drugs [21].

17-DMAG belongs to the category of drugs known as HSP90 inhibitors [10,11,12]. HSP90 is a chaperone protein that plays a crucial role in stabilizing and assisting in the correct folding of various client proteins, many of which are involved in cell growth, proliferation, and survival pathways [11]. The inhibition of HSP90 disrupts these essential cellular processes, making it a promising target for cancer therapies. The general anticancer effect of 17-DMAG lies in its ability to inhibit HSP90, which leads to the degradation of client proteins, ultimately leading to cell cycle arrest, apoptosis (programmed cell death), and tumor growth inhibition. In the context of gastric cancer, 17-DMAG has been observed to inhibit the proliferation of gastric cancer cells, induce apoptosis (programmed cell death) in these cells, reduce tumor growth in preclinical gastric cancer models, and enhance the sensitivity of gastric cancer cells to various chemotherapeutic agents [10,11,12,22].

17-DMAG does not exclusively target tumor cells, leading to the occurrence of several adverse effects when used as an anticancer therapy. The well-known adverse effects of 17-DMAG include gastrointestinal disturbances, fatigue, hepatotoxicity, hematological effects such as leukopenia and thrombocytopenia, electrocardiographic changes indicative of cardiotoxicity, neurological effects such as peripheral neuropathy or dizziness, and skin rash [10,11,12]. Some of the more severe side effects of 17-DMAG, such as liver toxicity and myelosuppression, may be life-threatening [11,23]. Consequently, vigilant monitoring of patients during 17-DMAG treatment is imperative. Therefore, the development of targeted therapies that selectively deliver 17-DMAG to tumor cells holds promise for enhancing its anticancer efficacy while mitigating its adverse effects. Our findings revealed that loading 17-DMAG into exosomes expressing proteins targeting MKN45 cells resulted in the highest degree of targeted delivery with the most pronounced antitumor effects, as evidenced by both in vitro and in vivo experiments.

The experimental results demonstrate that the drug-loaded targeted exosomes exhibited the most substantial therapeutic efficacy in both in vitro and in vivo experiments. Although this study did not assess adverse effects, it is anticipated that targeted delivery will augment the anticancer effect of the drug while concurrently diminishing adverse effects. Subsequent experiments will be conducted to elucidate this. The anticancer effects of 17-DMAG-loaded DE532 exosomes were comprehensively confirmed, including their effect on cell proliferation and alterations in apoptosis-related markers and antioxidant enzymes in a xenograft model, which resulted in reduced tumor size and weight. DE532 exosomes demonstrated a certain degree of anticancer effect, suggesting the involvement of MKN45 protein expressed on the surface of gastric cancer cells in the oncogenesis of gastric cancer [18]. These experimental results collectively highlight the application of drug delivery within genetically engineered exosomes as an efficacious anticancer approach. Given that various chemotherapeutic agents currently lack tumor-specific targetability, the use of targeted exosomes as vehicles has the potential to confer targetability to nonspecific chemotherapeutic agents.

Currently, this study has not been able to present an alternative for the development of exosome mass production and purification technology, and there is a lack of information on methods to improve the targeting for each type of cancer cell. Additionally, in the current study, experiments were conducted with a known target of a cancer cell line called DE532; however, its applicability to actual patient cancer cells may be difficult, acknowledging a need for the continued development of appropriate targets. Strategies to prevent autoimmune reactions caused by the excessive activation of the immune system by exosomes are also needed.

## 4. Material and Methods

### 4.1. Cell Culture

Human ASCs were obtained from Hurim BioCell Co. (Seoul, Republic of Korea) and cultured in DMEM/low glucose (GibcoBRL, Waltham, MA, USA) supplemented with antibiotics (penicillin–streptomycin; GibcoBRL) at 37 °C in a humidified atmosphere with 5% CO_2_ in an incubator. AGS and MKN45 cells were procured from the Korea Cell Line Bank (KCLB, Seoul, Republic of Korea). AGS and MKN45 gastric cancer cells were maintained in RPMI1640 (Hyclone, Logan, UT, USA) supplemented with 10% fetal bovine serum (Hyclone) and 1% penicillin/streptomycin (Gibco BRL, Waltham, CA, USA) at 37 °C in a humidified atmosphere with 5% CO_2_ in an incubator.

### 4.2. Generation of Genetically Engineered Exosomes

To generate tEx, the ASC cultures were grown to 90% confluence and subjected to starvation by removing fetal bovine serum for 24 h. After transfecting 4 µg of pDisplay-DE532 (MKN45 cell-targeted peptide: VETSQYFRGTLS(Val-Glu-Thr-Ser-Gln-Tyr-Phe-Arg-Gly-Thr-Leu-Ser)) into ASCs for 24 h, the conditioned medium was collected and centrifuged at 2500× *g* for 15 min at 4 °C to remove cell debris. Exosome isolation reagent (Thermo Fisher, Waltham, MA, USA) was then added to the conditioned medium, and the solution was incubated overnight at 4 °C. Exosomes were isolated from the conditioned medium using differential centrifugation at 10,000× *g* for 60 min at 4 °C. The precipitated exosomes were recovered through standard centrifugation at 10,000× *g* for 60 min, and the pellet was resuspended in phosphate-buffered saline (PBS).

### 4.3. Cargo Loading into the Exosomes

To investigate whether electroporation could effectively load cargo (17-DMAG) into exosomes, ASC exosomes were isolated using either total exosome isolation reagent or ultracentrifugation and then resuspended in an electroporation buffer. Electroporation was conducted using a NEPA 21 SUPER Electroporator with 0.2-cm cuvettes and aluminum electrodes. The samples, which had a volume of 100 µL (containing a maximum of 100 µg of total exosome protein), were electroporated at 200 V for 5 ms. Finally, the electroporated samples were analyzed using NTA and TEM, as previously described.

### 4.4. Flow Cytometry Analyses

Exosomes were suspended in ice-cold PBS (108 particles/mL) and incubated for 2 h at 4 °C with 20 μL of monoclonal antibodies(1:50): CD63-PE (BD Biosciences, Dubai, UAE, catalog number 556020), CD81-FITC (BD Bioscience, catalog number 555445), c-Myc-Alexa Fluor 488 (R&D systems, Minneapolis, MN, USA, catalog number IC3696G), and c-Myc-PE (R&D systems, catalog number IC3696P). Analysis was performed using the NxT Attune Flow Cytometer cell analyzer. All control samples were run along with the experimental samples, and all experiments were repeated three times.

### 4.5. NTA

NTA was performed using the ZetaView instrument (Particle Metrix GmbH, Ammersee, Bavaria, Germany). Each measurement involved the capture of 11 videos in one cycle. All measurements met the quality criteria of 50–150 particles per frame, 10^7^ particles/mL concentration, and valid tracks greater than 20%. The built-in ZetaView software (Version 8.05.16 SP3) was used to analyze the videos immediately after capture.

### 4.6. TEM

For TEM analysis, Ex and tEx suspended in PBS were placed on Formvar carbon-coated grids and left for 10 min. Subsequently, excess liquid was removed using filter paper. Negative staining was performed using a 1% uranyl acetate solution for 10 min, and any excess liquid was removed using filter paper. The grids were then left to dry at room temperature(25 °C). The adsorbed Ex and tEx were observed using a Hitachi transmission electron microscope (HT7800) operating at 80–100 kV.

### 4.7. Cell Viability Assay

In a 96-well plate, 1 × 10^4^ AGS and MKN45 gastric cancer cells were seeded and incubated for 24 h before treatment. The Ez-Cytox cell viability assay kit (Daeil Lab Service Co., Ltd., Seoul, Republic of Korea) was used to measure the cell viability after 48 h. Absorbance was measured at 450 nm using a Synergy HTX Multi-Mode Microplate Reader (BioTek Instruments, Inc., Winooski, VT, USA).

### 4.8. Detection of Mitochondrial Superoxide

AGS and MKN45 gastric cancer cells were stained with MitoSOX (Molecular Probes, Waltham, MA, USA) to detect the mitochondrial superoxide levels. The cells were kept in the dark at 37 °C for 10 min and then analyzed using a laser-scanning microscope (Eclipse TE300: Nikon, Tokyo, Japan).

### 4.9. Real-Time PCR

The total RNA was extracted from AGS, MKN45, and mouse tumor tissues using the TRIzol reagent (Invitrogen, Carlsbad, CA, USA). Reverse transcription was performed with 1 µg of RNA using the RT-premix kit (TOYOBO, Osaka, Japan), following the manufacturer’s instructions. SYBR Green real-time quantitative PCR was conducted using the following primers: mouse Bcl-xL forward 5′-AAC ATC CCA GCT TCA CAT AAC CCC-3′ and reverse 5′-GCG ACC CCA GTT TAC TCC ATC C-3′; mouse Bax forward 5′-CTGCAGAGGATGATTGCCG-3′ and reverse 5′-TGCCACTCGGAAAAAGACCT-3′; mouse GAPDH forward 5′-CGACTTCAACAGCAACTCCCACTCTTCC-3′ and reverse 5′-TGGGTGGTCCAGGGTTTCTTACTCCTT-3′. Reactions were performed using the Applied Biosystems StepOnePlus Real-Time PCR System (Thermo Fisher Scientific, Waltham, MA, USA).

### 4.10. Western Blot Analysis

AGS and MKN45 gastric cancer cells and mouse tissues were lysed using the EzRIPA Lysis kit (ATTO Corporation; Tokyo, Japan) and quantified using the Bradford reagent (Bio-Rad, Hercules, CA, USA). Proteins were visualized using Western blot analysis using primary antibodies (1:1000 dilution) from Cell Signaling Technology (Beverly, MA, USA) followed by horseradish peroxidase-conjugated secondary antibodies (1:2000 dilution) from Vector Laboratories (Burlingame, CA, USA). Specific immune complexes were detected using the Western Blotting Plus Chemiluminescence Reagent (Millipore, Bedford, MA, USA). Primary antibodies against Bcl-xL, BAX, and β-actin and horseradish peroxidase-conjugated secondary antibodies were obtained from Cell Signaling Technology (Beverly, MA, USA). The original Western blot images are available in the ‘Original Western blots’.

### 4.11. Immunohistochemical Analysis

For immunohistochemical analysis, formalin-fixed paraffin-embedded tissue sections were deparaffinized, rehydrated in an ethanol series, and subjected to epitope retrieval using standard procedures. Antibodies against Bcl-xL, BIM, SOD, and catalase were used for immunohistochemical staining (Cell Signaling Technology, Danvers, MA, USA). The samples were examined under a laser-scanning microscope (Eclipse TE300; Nikon, Tokyo, Japan) to analyze the expression of these antibodies.

### 4.12. Animals and Study Design

Five-week-old male BALB/c nude mice (Orient Bio, Seongnam, Republic of Korea) were used for the comparative modeling of subcutaneous tumor growth. AGS and MKN45 gastric cancer cells (5 × 10^6^) were injected subcutaneously into each mouse. All animal studies were conducted in compliance with the guidelines of the Institute for Laboratory Animal Research, Korea (IRB No: CUMC-2022-0141-01). The mice were weighed twice weekly. Detailed weights are provided in a Appendix A. Seven days after tumor cell injection, all the mice had measurable tumors. The in vivo efficacy was evaluated by randomly grouping the mice (n = 5 per group) and treating them intravenously with PBS (control), Ex (10^10^ Ex particles in 100 µL of PBS, twice a week), tEx (10^10^ Ex particles in 100 µL of PBS, twice a week), Ex (17-DMAG, 2.5 mg/mL; 10^10^ Ex particles in 100 µL of PBS, twice a week), or tEx (17-DMAG, 2.5 mg/mL; 10^10^ Ex particles in 100 µL of PBS, twice a week) for 30 days. Ex and tEx were labeled using the ExoGlowTM Ex Labeling Kit (SBI Biosciences, Palo Alto, CA, USA). ExoGlow-Vivo-labeled Ex and tEx showed robust signals in vivo. The tumor size was measured twice a week using a caliper, and the tumor volume (V) was calculated using the following formula: length × width^2^ × 0.5236. After treatment completion, all mice were euthanized. The mice were anesthetized for 5 min using a gas mixture of inhaled oxygen (3 mmHg) and nitrous oxide (7 mmHg), which was administered through a specially designed chamber. After the experiment was completed, the mice were euthanized with 100% carbon dioxide gas for 5 min to obtain tissue samples.

### 4.13. Statistical Analysis

All data were analyzed using SPSS software (version 11.0; SPSS Inc., Chicago, IL, USA) and presented as the mean ± standard deviation. Statistical comparisons among the groups were performed using the Kruskal–Wallis test. Probability values of *p* < 0.05 were considered statistically significant.

## 5. Conclusions

Our study demonstrates that 17-DMAG-loaded DE532 exosomes, denoted as tEx(D), exhibited superior targetability and anticancer efficacy against gastric cancer. These findings validate the potential of using genetically engineered exosomes as drug delivery systems, offering enhanced therapeutic outcomes while minimizing adverse effects. This study paves the way for further investigation of the clinical applications of 17-DMAG-loaded DE532 exosomes, which hold promise for improving the management of gastric cancer. Additionally, tEx (without cargo) showed considerable antitumor effects. However, the mechanism underlying this effect needs to be elucidated further.

## Figures and Tables

**Figure 1 ijms-25-08762-f001:**
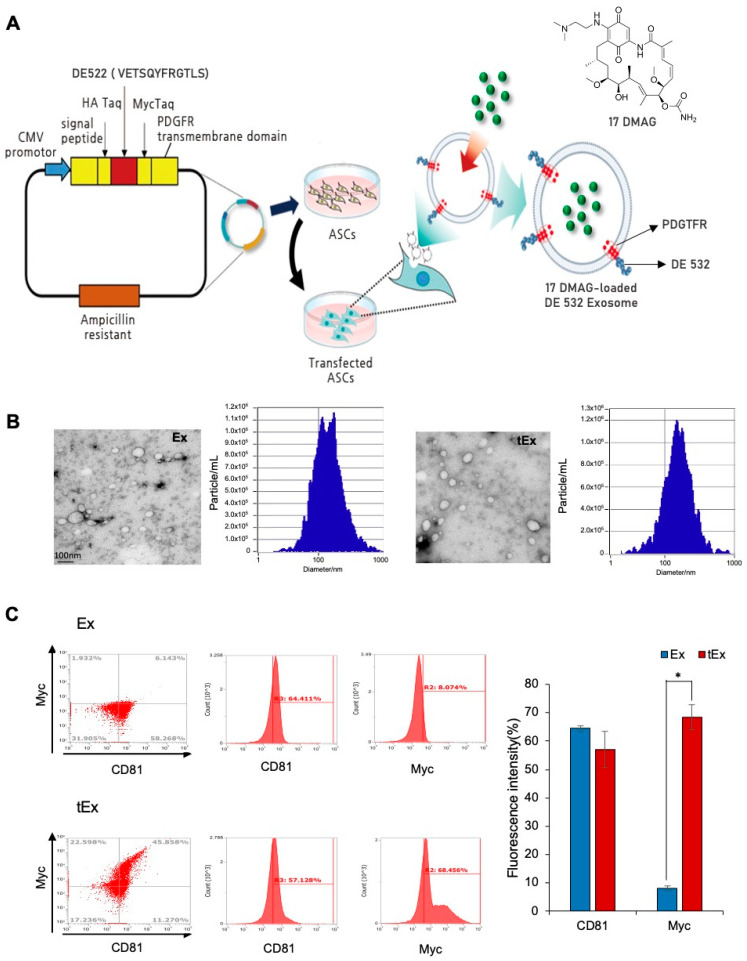
Generation and characterization of 17DMAG-loaded DE532 exosomes. (**A**) Schematic representation of the process for engineering 17DMAG-loaded DE532 exosomes. (**B**) Transmission electron microscopy (TEM) images comparing the morphology of exosomes obtained from non-transfected ASCs (Ex) and exosomes obtained from transfected ASCs (tEx). No apparent visual differences in the nanoparticle-like structures were observed between the two groups. (**C**) Flow cytometry analysis comparing the expression of the exosome marker CD81 and the Myc marker (indicative of DE532 peptide expression) in Ex and tEx. CD81 was expressed at 64.2% in Ex and 53.4% in tEx, while the Myc marker was expressed at 8.1% in Ex and 63.6% in tEx, confirming that tEx possess exosome characteristics and carry the DE532 peptides. Note: * *p* < 0.05.

**Figure 2 ijms-25-08762-f002:**
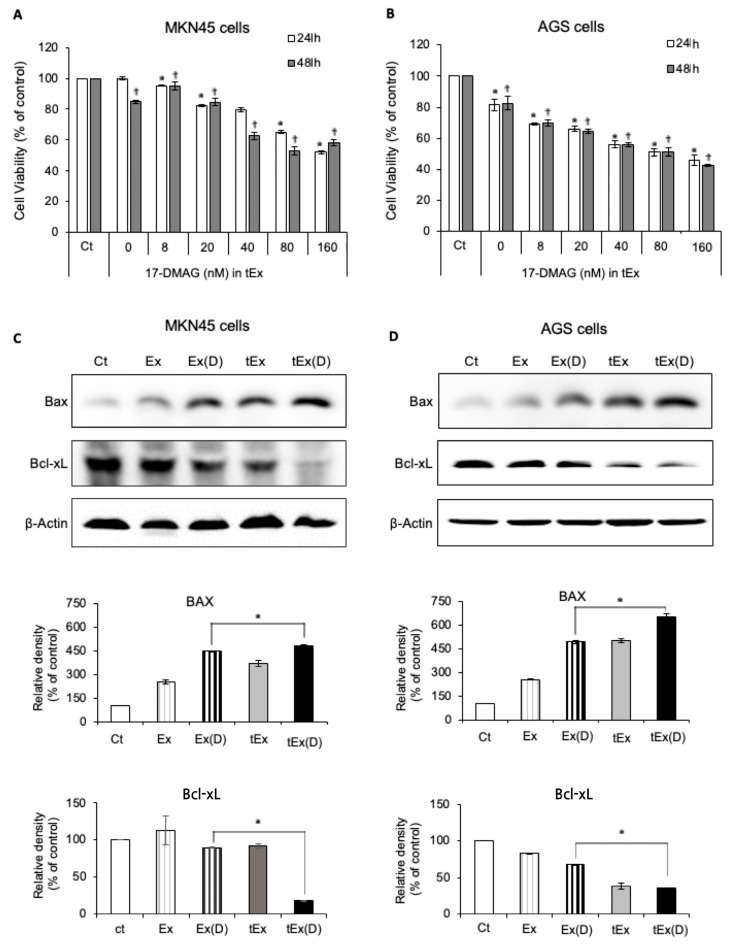
In vitro antitumor effects of 17DMAG-loaded DE532 exosomes. MTT assay results demonstrating the concentration- and time-dependent decrease in the cell viability of MKN45 (**A**) and AGS (**B**) cells treated with tEx(D) encapsulating varying concentrations of 17-DMAG. Western blot analysis of apoptotic markers (BAX and Bcl-xL) in MKN45 (**C**) and AGS (**D**) cells treated with Ex, Ex(D), tEx, and tEx(D) containing a total of 5 × 10^6^ exosome particles, with a concentration of encapsulated 17-DMAG at 40 nM. Treatment with tEx(D) resulted in the most significant increase in the expression of pro-apoptotic marker BAX and the greatest decrease in the expression of anti-apoptotic marker Bcl-xL (*p* < 0.05). Western blots were prepared from the same sample and divided onto separate membranes, each probed with a different antibody. Finally, the results were quantified using Image Lab™ software (http://imagej.nih.gov/ij/download.html). The band intensities were calculated using ImageJ 1.54d software. GAPDH was used as an internal control for the total protein measurement. Note: * *p* < 0.05. † *p* < 0.05.

**Figure 3 ijms-25-08762-f003:**
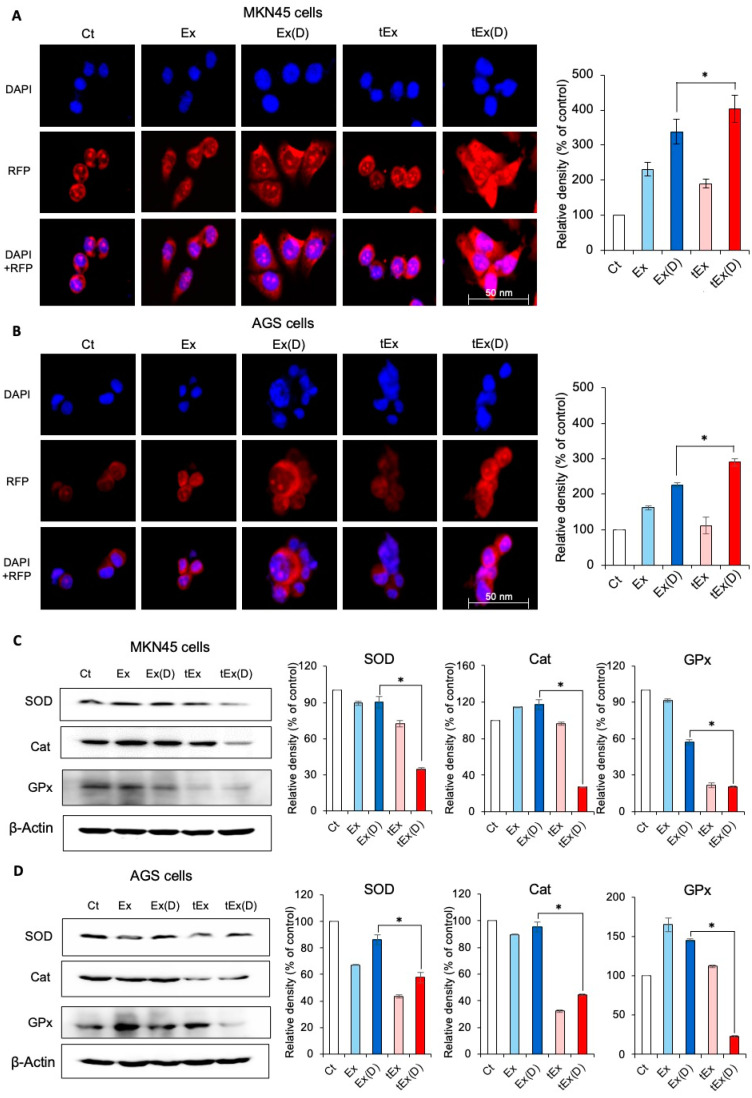
Effects of 17DMAG-loaded DE532 exosomes on mitochondrial ROS and antioxidant enzymes. MitoSOX immunofluorescence results comparing changes in mitochondrial reactive oxygen species (ROS) in MKN45 (**A**) and AGS (**B**) cells treated with Ex, Ex(D), tEx, and tEx(D) exosomes. Both the Ex(D) and tEx(D) groups exhibited increased MitoSOX expression compared to the control group, with the tEx(D) group showing the most significant increase in MitoSOX expression. Western blot analysis of antioxidant enzyme expression in MKN45 (**C**) and AGS (**D**) cells after treatment with Ex, Ex(D), tEx, and tEx(D) exosomes. Treatment with tEx(D) resulted in the lowest expression of antioxidant enzymes among all groups. Western blots were prepared from the same sample and divided onto separate membranes, each probed with a different antibody. Finally, the results were quantified using Image Lab™ software (Bio-Rad). The band intensities were calculated using ImageJ 1.54d software. GAPDH was used as an internal control for the total protein measurement. Note: * *p* < 0.05.

**Figure 4 ijms-25-08762-f004:**
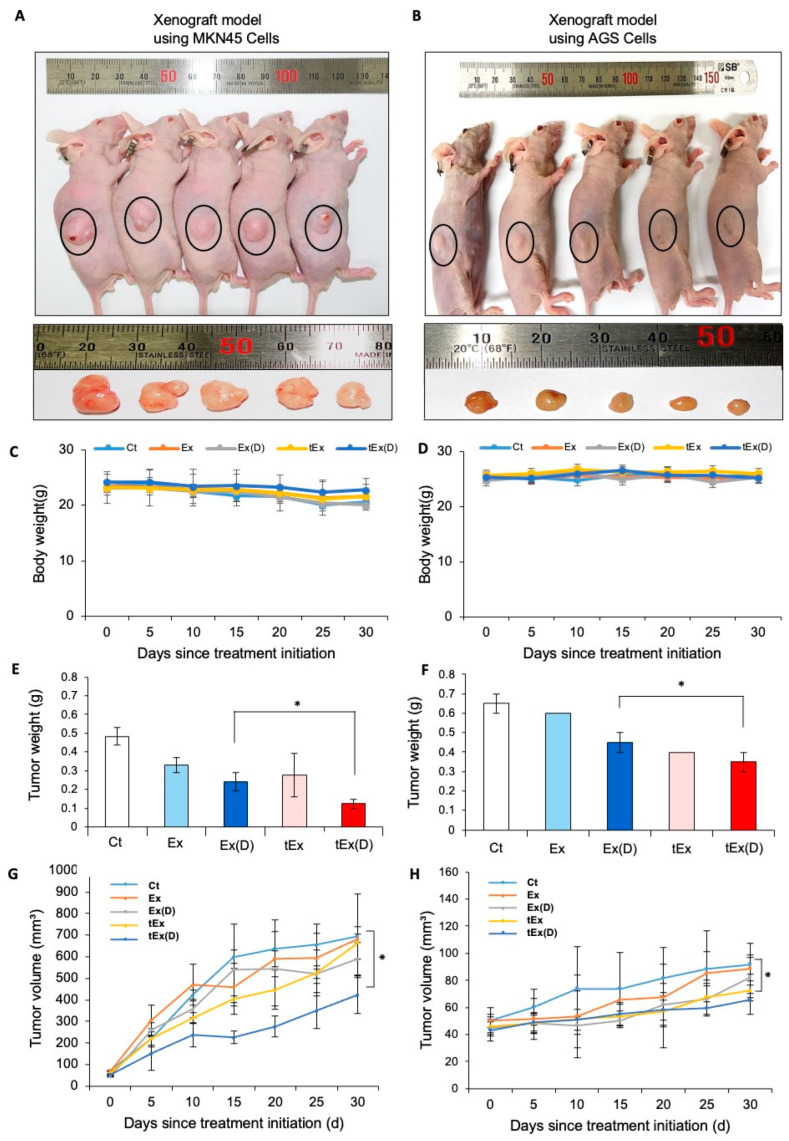
In vivo anticancer effects of 17DMAG-loaded DE532 exosomes. Comparison of tumor sizes (**A**,**B**), average mice weight (**C**,**D**), tumor weights (**E**,**F**), and tumor volume over time (**G**,**H**), showing that the tEx(D) group exhibited the smallest values among all groups in the gastric cancer xenograft mouse models generated by MKN45 (**A**) and AGS (**B**) cells. black circle: The location where cells are implanted. Note: * *p* < 0.05.

**Figure 5 ijms-25-08762-f005:**
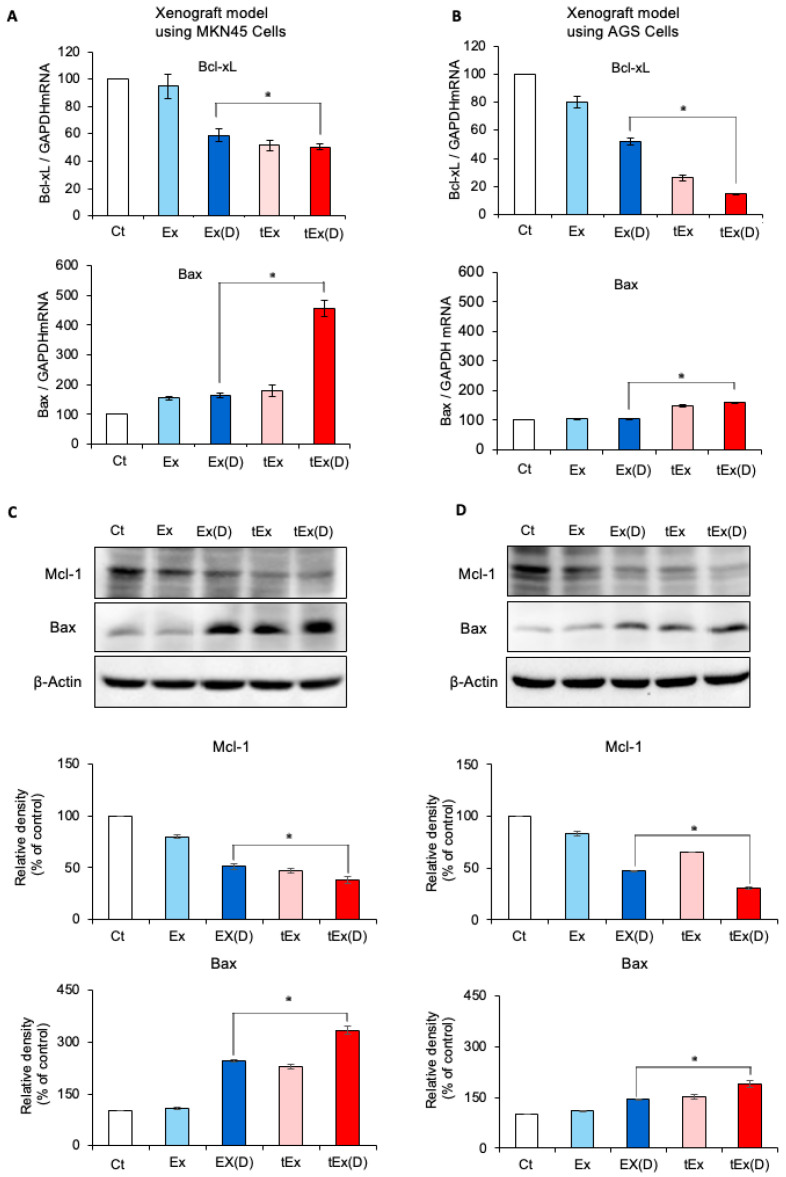
Real-time PCR and Western blotting determined the expressions of apoptosis-related markers in the tumor tissues of mouse xenograft models. Real-time PCR analysis of apoptosis-related markers in the MKN45 (**A**) and AGS (**B**) cell xenograft models. Treatment with tEx(D) resulted in significant upregulation of the pro-apoptotic marker Bax and downregulation of the anti-apoptotic marker Bcl-xL to the greatest extent. Western blot analysis of apoptosis-related proteins in the MKN45 (**C**) and AGS (**D**) cell xenograft models. Among all treatment groups, the tEx(D) treatment significantly upregulated the expression of the pro-apoptotic marker Bax and downregulated the anti-apoptotic marker Bcl-xL. Western blots can be prepared from the same sample and divided onto separate membranes, each probed with a different antibody. Finally, the results were quantified using Image Lab™ software (Bio-Rad). The band intensities were calculated using ImageJ 1.54d software. GAPDH was used as an internal control for the total protein measurement. Note: * *p* < 0.05.

**Figure 6 ijms-25-08762-f006:**
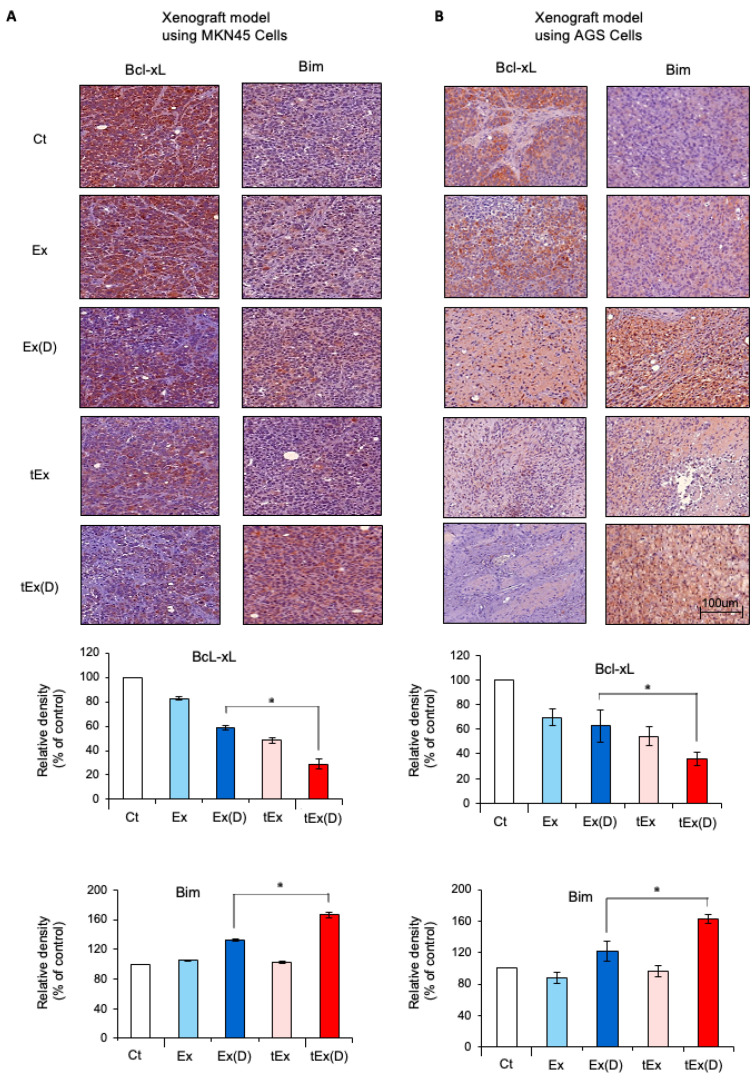
Immunohistochemical stains showing expressions of apoptosis-related markers. Immunohistochemical staining of tumor tissues obtained from the MKN45 (**A**) and AGS (**B**) cell xenograft models demonstrated that the tEx(D) treatment resulted in the most significant increase in the expression of the pro-apoptotic marker BIM and the most substantial decrease in the expression of the anti-apoptotic marker Bcl-xL. Note: * *p* < 0.05.

**Figure 7 ijms-25-08762-f007:**
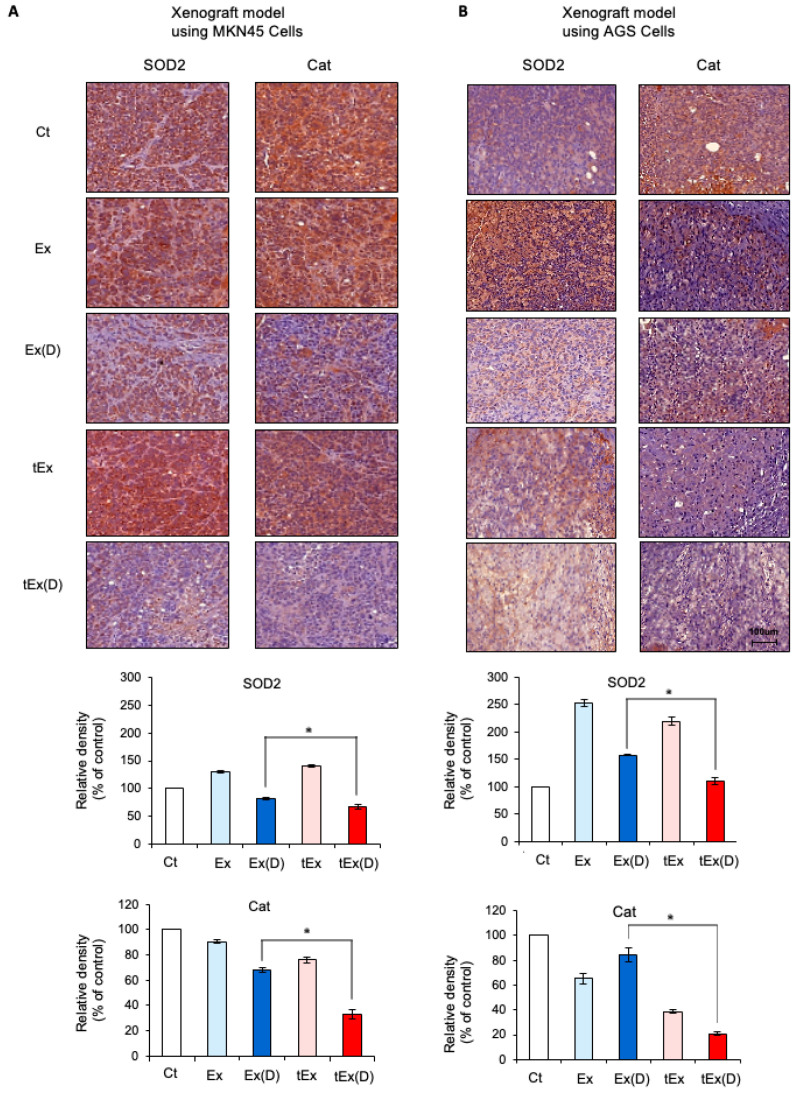
Immunohistochemical stains showing expressions of antioxidant enzymes. Immunohistochemical staining of tumor tissues obtained from the MKN45 (**A**) and AGS (**B**) cell xenograft models demonstrated that the tEx(D) treatment led to the most significant reductions in the expressions of SOD2 and catalase. Note: * *p* < 0.05.

## Data Availability

The current study datasets are original data from our laboratory and are not available online.

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
