# Peer review of "Enhanced Efficacy of Gastric Cancer Treatment through Targeted Exosome Delivery of 17-DMAG Anticancer Agent"

_ijms, 2024, doi:10.3390/ijms25168762_

Round 1
Reviewer 1 Report
Comments and Suggestions for Authors
The experiment design is well thought out and I recommend the paper for publication after the major revision of the following point in the result section.
paragraph 2.2: what the authors mean with "tEx loaded with 50nM 17-DMAG"? 1) Is this the 17-DMAG concentration before electroporation? 2) Have the exosome been isolated from the free 17-DMAG?
In the first hypothesis, has been the number of vesicles maintained constant at various concentrations of 17-DMAG (Fig. 2A-B)? In this case a control with free 17-DMAG (without exosomes) is missing (I suppose that the control that appears in the figures is just the medium (or PBS), free of exosomes. Is it right?) and it would be useful indicate the amount of free 17-DMAG that is considered toxic for cells and living organisms highlighting a comparison with the amount administered by exosomial formulation.
In the second hypothesis, how has been the cargo quantified?
Finally, at the beginning it is written that exosomes were loaded with 50 nM of 17-DMAG, at the end of the paragraph it is reported 40 nM (not nm), and in Fig. 2A-B there is a study at different concentrations of 17-DMAG. Maybe is 40-50 nM the efficient concentration highlighted by the study in function of concentration?
Other minor comments/detection of mistakes are the following:
RESULTS
paragraph 2.1 line 6: electroporation
paragraph 2.3: I suppose that RFP refers to MitoSOX signal (red color). Maybe it could be specified better.
Fig.1: the resolution of the histograms should be increased (similarly to the last histogram in panel C): the scales are illegible and it is unclear what represents the first histogram in panel B.
Fig.5: what is Mcl-1? Maybe it is a mistake.
DISCUSSION
line 16: expression (not suppression)
CONCLUSION
Another conclusion seems that also tEx (without cargo) have an antitumoral effect, so could be also interesting to investigate this point.
M&M
paragraph 4.2: centrifugation at 10000 g to recover exosomes around 150 nm in diameter appears a bit low. Tipically at this acceleration the microvesicles precipitate and just a residual amount of exosomes (tipically the small ones). Maybe using higher acceleration could increase the amount of isolated vesicles. If it is a well established protocol for the isolation of these vesicles, the authors should insert the reference.
Author Response
Reviewer 1.
The experiment design is well thought out and I recommend the paper for publication after the major revision of the following point in the result section.
paragraph 2.2: what the authors mean with "tEx loaded with 50nM 17-DMAG"?
1) Is this the 17-DMAG concentration before electroporation?
Answer> Yes! 50nM is the concentration of 17-DMAG before electoroporation.
2) Have the exosome been isolated from the free 17-DMAG? In the first hypothesis, has been the number of vesicles maintained constant at various concentrations of 17-DMAG (Fig. 2A-B)?.
Answer> Yes, we washout the unloaded 17-DMAG (300Da) using 10k amicon filter after electroporation. In addition, we used same amount of vesicle for cancer cell treatment
In this case a control with free 17-DMAG (without exosomes) is missing (I suppose that the control that appears in the figures is just the medium (or PBS), free of exosomes. Is it right?) and it would be useful indicate the amount of free 17-DMAG that is considered toxic for cells and living organisms highlighting a comparison with the amount administered by exosomial formulation.
Answer> Yes, control is only medium. We would like to show the effect of target exosome compared with non-targeted exosome only. In addition, in the previous study, we showed the effect of 17-DMAG at anti-proliferative effect on AGS cell. (Oncotarget 2017, 8, 56473-56489.) comparing to the previous study, targeted exosome require less concentration of 17-DMAG than unloaded 17-DMAG. Although we don’t have sufficient data ragarding the toxic dose of 17-DMAG, some clincal effects were described in the discussion section with reference.
In the second hypothesis, how has been the cargo quantified?
Answer> After washout the unloaded 17-DMAG, amount of 17-DMAG was checked with HPLC after breakdown the exosome membrane with 1% triton.
Finally, at the beginning it is written that exosomes were loaded with 50 nM of 17-DMAG, at the end of the paragraph it is reported 40 nM (not nm), and in Fig. 2A-B there is a study at different concentrations of 17-DMAG. Maybe is 40-50 nM the efficient concentration highlighted by the study in function of concentration?
Answer> Electroporation underwent with 50nM of 17-DMAG, 40nM was the concentration of 17-DMAG which is loaded in the exosomes and main experiments were performed with 40nM exosome loaded 17-DMAG.
Other minor comments/detection of mistakes are the following:
RESULTS paragraph 2.1 line 6: electroporation
paragraph 2.3: I suppose that RFP refers to MitoSOX signal (red color). Maybe it could be specified better.
Answer> Yes, we can identify red signal treated with MitoSOX.
Fig.1: the resolution of the histograms should be increased (similarly to the last histogram in panel C): the scales are illegible and it is unclear what represents the first histogram in panel B
Answer> We will revise the figure quality.
Fig.5: what is Mcl-1? Maybe it is a mistake.
Answer> No it is not a mistake. Myeloid leukemia 1 (MCL-1) is an anti-apoptotic protein that inhibits cell death and promotes cell survival. In gastric cancer, higher expression levels of MCL-1 are associated with poor patient prognosis and may lead to resistance.
DISCUSSION line 16: expression (not suppression)
Answer> we revised the manuscript.
CONCLUSION Another conclusion seems that also tEx (without cargo) have an antitumoral effect, so could be also interesting to investigate this point.
Answer> Yes! It was also interesting point that tEx(without cargo) also showed the anti-cancer effect.
We would like to revise the conclusion adding this point.
“In addition, tEx(without cargo) also showed considerable antitumoral effect. The mechanism of this effect should be elucidated more.
M&M paragraph 4.2: centrifugation at 10000 g to recover exosomes around 150 nm in diameter appears a bit low. Tipically at this acceleration the microvesicles precipitate and just a residual amount of exosomes (tipically the small ones). Maybe using higher acceleration could increase the amount of isolated vesicles. If it is a well established protocol for the isolation of these vesicles, the authors should insert the reference.
Answer> The exosome extraction method used by this researcher involved the Thermo Fisher exosome isolation kit. This kit utilizes the immunoprecipitation method for exosome protein extraction and is used according to the kit protocol.
Reviewer 2 Report
Comments and Suggestions for Authors
Comments:
1. In the introduction section, it is necessary to properly introduce exosomes (extracellular vesicles), including their different types, their primary structure and composition, and the reasons why they encompass biocompatibility, low immunogenicity, and an intrinsic capacity for transporting biomolecules. Providing this information here will greatly help readers who are not familiar with them to understand. I would suggest authors to cite this manuscript https://www.mdpi.com/2227-9059/9/10/1373
2. In the described function of 17-DMAG, it is important to specify which cellular processes it disrupts. Additionally, at which stage does it exert its inhibitory effects to orchestrate cell cycle arrest?
3. Authors should discuss here about some cytotoxic chemotherapy agents and their mechanisms of action that have been previously used for this cancer, and attention should also be focused on their drawbacks.
4. The methods section could have been improved by providing a brief description of the reasoning behind the methods chosen.
5. The manuscript is lacking figures; there are only figure legends, which are making it quite difficult to understand the results section. Although, 7 figure legends are here.
6. It is necessary to compare the previous results of studies of gastric cancer treatment with 17-DMAG drugs in the discussion section here.
7. I would suggest to include limitation of this study.
Comments on the Quality of English LanguageMinor English editing is required.
Author Response
- In the introduction section, it is necessary to properly introduce exosomes (extracellular vesicles), including their different types, their primary structure and composition, and the reasons why they encompass biocompatibility, low immunogenicity, and an intrinsic capacity for transporting biomolecules. Providing this information here will greatly help readers who are not familiar with them to understand. I would suggest authors to cite this manuscript https://www.mdpi.com/2227-9059/9/10/1373
Answer> Thank you for comment. We will revise the introduction to deliver general information about exosome and additionally cite mentioned like followings.
Exosomes are natural nano-sized vesicles secreted by cells and vary in size from 30 to 150 nm. They can transport various biomolecules and play an important role in interactions with the immune system, making them highly suitable for precise drug delivery (ref. Biomedicines 2021, 9, 1373. https://doi.org/10.3390/biomedicines9101373 Extracellular Vesicle-Based Therapy for COVID-19:Promises, Challenges and Future Prospects)
- In the described function of 17-DMAG, it is important to specify which cellular processes it disrupts. Additionally, at which stage does it exert its inhibitory effects to orchestrate cell cycle arrest?
Answer> We revised the sentence in 17-DMAG section like followings adding relevant reference.
By inhibiting HSP90, 17-DMAG disrupts G2/M cellular proliferation cycle, promising profound implications for cancer therapy.( Cell Stress Chaperones. 2016 Mar; 21(2): 339–348.)
- Authors should discuss here about some cytotoxic chemotherapy agents and their mechanisms of action that have been previously used for this cancer, and attention should also be focused on their drawbacks.
Answer> Thank you for comment. It would be helpful to revise the discussion as you mentioned. For that, we added the contens regarding the cytotoxic agents for gastric cancer like followings
“The need for new anticancer treatments arises from the limitations of existing therapies. Cytotoxic chemotherapy agents such as 5-FU, Cisplatin, and Oxaliplatin are widely used to treat stomach cancer due to their effectiveness in attacking cancer cells. However, these treatments also affect normal cells, leading to various side effects.
Targeted treatments have been developed to alleviate some of these side effects, but problems remain. For example, Trastuzumab for HER2-positive gastric cancer and Ramucirumab for advanced gastric cancer have shown improved results compared to traditional chemotherapy. However, these treatments have limitations. They are not applicable to all patients, are expensive, and, like other cytotoxic anticancer drugs, there is the issue of resistance that must be considered.”
Below paragraph was removed for the natural flow of the discussion
“Targeting MKN45 as a surface marker on gastric cancer cells holds considerable promise, given its potential contribution to diagnosis and treatment strategies. Studies have reported that MKN45 is highly expressed in up to 80% of gastric cancer cells, making it a promising target for cancer-specific interventions [14-16]. Furthermore, MKN45 is also expressed in various other tumor types, such as liver (90%), gastric (75%), and pancreatic (70%) cancers, indicating its potential relevance across a wide range of cancers [14]. By contrast, MKN45 expression is relatively low or absent in healthy cells, with only about 5% of normal cells expressing MKN45 at detectable levels [14]. In summary, focusing on MKN45 as a surface marker for gastric cancer cells has considerable potential due to its high expression in cancerous cells and limited presence in normal cells, offering a valuable strategy for targeted diagnosis and treatment.
“
- The methods section could have been improved by providing a brief description of the reasoning behind the methods chosen.
Answer> Thank you for comment.
However, the necessity for using exosome was described in introduction already.
- The manuscript is lacking figures; there are only figure legends, which are making it quite difficult to understand the results section. Although, 7 figure legends are here.
Answer> There were some problems in uploading the figures.
Now all figures are on the right place.
- It is necessary to compare the previous results of studies of gastric cancer treatment with 17-DMAG drugs in the discussion section here.
Answer> In the discussion, our previous study regarding the anti-tumor effect of 17-DMAG not with exosome was described.
- I would suggest to include limitation of this study.
Answer> Thank you for comment. We will add the limitation of our study like followings.
“Currently, this study has not been able to present an alternative to the development of exosome mass production and purification tech-nology, and there is a lack of information on methods to improve tar-geting for each type of cancer cell. In the current study, experiments were conducted with a known target of a cancer cell line called DE532, but applicability to actual patient cancer cells may be difficult and it would be good to also present the need for continued development of appropriate targets. In the future, strategies to prevent autoimmune reactions caused by excessive activation of the immune system by exosomes are needed.”
Reviewer 3 Report
Comments and Suggestions for Authors
The manuscript entitled "Precision Drug Delivery: MKN45-targeted exosomes deliver 17-DMAG for gastric cancer in mouse models" by Say-June Kim et al., was submitted to IJMS as article for possible consideration. However, the figures and tables were invisible. I checked up the enclosed gels and membranes. But the Tables and Figures were suggested to re-provided.
Looking forward to reformatted manuscript with the results' table and figure.
Author Response
The manuscript entitled "Precision Drug Delivery: MKN45-targeted exosomes deliver 17-DMAG for gastric cancer in mouse models" by Say-June Kim et al., was submitted to IJMS as article for possible consideration. However, the figures and tables were invisible. I checked up the enclosed gels and membranes. But the Tables and Figures were suggested to re-provided.
Looking forward to reformatted manuscript with the results' table and figure.
Answer> All figures have been uploaded with manuscript now. Sorry for the inconvenience.
Reviewer 4 Report
Comments and Suggestions for Authors
Authors have addressed the use of exosomes delivering 17-DMAG drug to the gastric cancer model in mice. Authors have designed the experiments nicely and are appropriate to justify the results. However, there are a few points that authors need to work on.
1. What is the size of the exosomes for both transfected and non-transfected exosomes?
2. Can authors make any suggestions for the localization of these exosomes in the tumor-bearing mice? Where the drug delivered was observed, it's important to mention as the authors have highlighted this in their manuscript title.
Comments on the Quality of English Language
English is ok but can be improved a little.
Author Response
Reviewer 4
Authors have addressed the use of exosomes delivering 17-DMAG drug to the gastric cancer model in mice. Authors have designed the experiments nicely and are appropriate to justify the results. However, there are a few points that authors need to work on.
What is the size of the exosomes for both transfected and non-transfected exosomes?
Answer> Thank you for asking. Regardless of whether they have been transfected or not, the size of exosomes is generally in the nano-size range, and transfection itself does not have a significant effect on this size. The size of the exosomes extracted by this research team was 100 nm for both control exosomes and targeted exosomes.
Can authors make any suggestions for the localization of these exosomes in the tumor-bearing mice?
Where the drug delivered was observed, it's important to mention as the authors have highlighted this in their manuscript title.
Answer> It was very important and critical review.
To prove the drug distribution, we must perform additional animal experiments with tEx(D) marked with fluorescent. There needs more time to complete additional experiments until the due date of revision. We have already started our additional experiments. We might upgrade our manuscript on the 2nd round of revision.
Round 2
Reviewer 3 Report
Comments and Suggestions for Authors
Major revision
A manuscript entitled “Precision Drug Delivery: MKN45-targeted exosomes deliver 17-DMAG for gastric cancer in mouse models” by Jung Hyun Park et al. was submitted as research article for possible consideration of publications on IJMS. The authors proposed that 17DMAG-loaded MKN45-targeting exosomes effectively delivered 17-DMAG to gastric cancer cells, and presented some enhanced anti-tumor effects. In general, it was an interesting study and the experiments were well designed. Since there was no line numbers in the manuscript, it is truly difficult to precisely mention or indicate the errs and all issues. So, before the current submission could be process a little bit further, in my mind, quite some issues should be carefully address, as follows,
1) The title was vague. It could be more specific. Precision Drug Delivery can be deleted because it looked like a review title. So, the original title of the manuscript should be modified to “17DMAG-loaded DE532 exosomes for specific delivery of 17-DMAG to target gastric cancer in mouse xenograft models” or another suitable title or a much better one.
2) Page 2,“Exosomes, being naturally occurring nanoscale vesicles secreted by cells,…” was repeated after the first sentence of the introduction.
3) Page 2, the sentence of “MKN45 is one of the most commonly used gastric cancer cell line and it express many biomarkers related with tumor progression. [16-18].” should be revised to “MKN45 is one of the most used gastric cancer cell lines and it expresses many biomarkers related with tumor progression [16-18].”. please avoid the typos and grammar errs.
4) In the introduction section, only MKN45 gastric cancer cell line was described but AGS cancer cell line was not compared even mentioned. Why?
5) In single one paragraph, MKN-45 and MKN45 were mixed and used? Why? Were they different?
6) Regarding the Fig. 1B, I am curious about the mean diameter of Exo was 151.5 nm or not. The same question remained for tExo.
7) There was a small thing. I am wondering, did Ex exactly mean Exo, and tEx mean tExo? As authors showed the terms differently in Fig.1C and in the manuscript.
8) Fig.2 should be revised. Bcl-Xl was used. And BCL-xL was used. What’s the difference then? In the main texts and in Fig. 5, Bcl-xL was used instead over there. Why? Why there were so many variants?
9) Figure 4 B, the size of the xenograft mouse, the middle one was smaller than the other four mouse. Did authors checkup the body weight of the treated xenograft mouse?
10) 4. Material and Methods 4.1. Cell culture In this part, the human adipose-derived stem cells (ASCs) were used but their culture condition was not clear.
11) 4.7. Cell viability assay “In a 96-well plate, 1 × 104 AGS” can be revised.
12) 4.12. Animals and study design authors said that “Five-week-old male BALB/c nude mice (Orient Bio, Seongnam, Republic of Korea) were used for comparative modeling of subcutaneous tumor growth.” And “The mice were weighed twice a week”. But the average weights of the mice were not reported.
13) GSH levels were not mentioned. But CAT and GPx and SOD2 were analyzed.
14) Open questions: The effects on redox systems in cells and cancer cells? Are there any effects of Exo on TXNRD1/2 in the gastric cancers in vivo and in vitro? side effects of ASCs Exo?Why and How did authors choose the cellular targets for RT qPCR detections? how did authors rule out other regulated cell death pathways?
15) BTW, a high-quality graphical abstract was missing. Good to have one.
Comments on the Quality of English LanguageEnglish language of the manuscript should be double-checked.
Author Response
A manuscript entitled “Precision Drug Delivery: MKN45-targeted exosomes deliver 17-DMAG for gastric cancer in mouse models” by Jung Hyun Park et al. was submitted as research article for possible consideration of publications on IJMS. The authors proposed that 17DMAG-loaded MKN45-targeting exosomes effectively delivered 17-DMAG to gastric cancer cells, and presented some enhanced anti-tumor effects. In general, it was an interesting study and the experiments were well designed. Since there was no line numbers in the manuscript, it is truly difficult to precisely mention or indicate the errs and all issues. So, before the current submission could be process a little bit further, in my mind, quite some issues should be carefully address, as follows,
1) The title was vague. It could be more specific. Precision Drug Delivery can be deleted because it looked like a review title. So, the original title of the manuscript should be modified to “17DMAG-loaded DE532 exosomes for specific delivery of 17-DMAG to target gastric cancer in mouse xenograft models” or another suitable title or a much better one.
-> Thank you for the valuable suggestion. We will revise the title as your recommendation like following.
“Enhanced Efficacy of Gastric Cancer Treatment through Targeted Exosome Delivery of 17-DMAG Anti-Cancer Agent”
2) Page 2,“Exosomes, being naturally occurring nanoscale vesicles secreted by cells,…” was repeated after the first sentence of the introduction.
->Thank you for your comment. We deleted the sentence.
3) Page 2, the sentence of “MKN45 is one of the most commonly used gastric cancer cell line and it express many biomarkers related with tumor progression. [16-18].” should be revised to “MKN45 is one of the most used gastric cancer cell lines and it expresses many biomarkers related with tumor progression [16-18].”. please avoid the typos and grammar errs.
-> Thank you. We revised as you recommended.
4) In the introduction section, only MKN45 gastric cancer cell line was described but AGS cancer cell line was not compared even mentioned. Why?
-> In the introduction, MKN45 was mainly discussed because the DE542 peptide was first developed from the phage display on MKN45. However, experiments underwent also on AGS cell-line to find the targetability on the other gastric cancer cell-line. We added following sentence in the last paragraph of introduction section.
“In addition, same experiments underwent with AGS cell-line along with MKN45 to explore the efficancy on the other gastric cancer cell-ine.”
5) In single one paragraph, MKN-45 and MKN45 were mixed and used? Why? Were they different?
Thank you for your comment. We revised all MKN-45 to MKN45.
6) Regarding the Fig. 1B, I am curious about the mean diameter of Exo was 151.5 nm or not. The same question remained for tExo.
Answer> Thank you for your question.
The research team extracted Ex and tEx and analyzed the particle size using the nanoparticle tracking analysis (NTA) equipment, Zetaview.
As a result, the main size of Ex was measured to be 141nm, and tEx was measured to be 148nm. Consequently, it was confirmed that there was no significant difference in the size of the two exosomes.
7) There was a small thing. I am wondering, did Ex exactly mean Exo, and tEx mean tExo? As authors showed the terms differently in Fig.1C and in the manuscript.
Answer> Thank you for your comment. We revised all “Exo” and “tExo” to “Ex” and “tEx”, respectively.
8) Fig.2 should be revised. Bcl-Xl was used. And BCL-xL was used. What’s the difference then? In the main texts and in Fig. 5, Bcl-xL was used instead over there. Why? Why there were so many variants?
Answer> Thank you for your comment. We revised Bcl-Xl and BCL-xL to Bcl-xL.
9) Figure 4 B, the size of the xenograft mouse, the middle one was smaller than the other four mouse. Did authors checkup the body weight of the treated xenograft mouse?
Answer> Yes, we checked the weight of the mouse during the experiment. 5 mice were used for each experiment. We will attach the average and standard deviation of weight change.
10) 4. Material and Methods 4.1. Cell culture In this part, the human adipose-derived stem cells (ASCs) were used but their culture condition was not clear.
Answer> Thank you for your comment. However, our other studies using the same culture method were published with similar contents. Can you comment more precisely for us to revise our manuscript.
11) 4.7. Cell viability assay “In a 96-well plate, 1 × 104 AGS” can be revised.
-> Thank you. We revised to “1 × 104 “
12) 4.12. Animals and study design authors said that “Five-week-old male BALB/c nude mice (Orient Bio, Seongnam, Republic of Korea) were used for comparative modeling of subcutaneous tumor growth.” And “The mice were weighed twice a week”. But the average weights of the mice were not reported.
-> We estimated mice weight serially. We will attach the average weight of the mice with supplementary file with the following revision.
“Average weight will be delivered with supplement file.”
13) GSH levels were not mentioned. But CAT and GPx and SOD2 were analyzed.
Answer> Thank you for your comment. GSH neutralizes reactive oxygen species (ROS) by converting to oxidized GSSG, but this study did not measure GSH. Instead, GPx was measured. GPx catalyzes the reaction H2O2 + 2GSH → GSSG + 2H2O, oxidizing GSH and removing H2O2.
In the tExo treated group, a decrease in GPx was observed, which indicates an increase in ROS within cancer cells, potentially leading to increased apoptosis.
14) Open questions: The effects on redox systems in cells and cancer cells? Are there any effects of Exo on TXNRD1/2 in the gastric cancers in vivo and in vitro? side effects of ASCs Exo?Why and How did authors choose the cellular targets for RT qPCR detections? how did authors rule out other regulated cell death pathways?
Answer> In both normal and cancer cells, the redox system plays a crucial role in responding to oxidative stress, cell survival, proliferation, and tumor formation. The redox system regulates reactive oxygen species (ROS), maintains intracellular antioxidant defenses, and influences various signaling pathways. In normal cells, the redox system is essential for maintaining normal cell function and survival. In cancer cells, alterations in the redox system can contribute to tumor survival and growth. Increased expression of antioxidant enzymes such as GSH, CAT, GPx, SOD, and TXNRD protects cancer cells from ROS. Therefore, understanding the redox system can provide new approaches to cancer treatment.
Although we did not directly measure TXNRD1/2, we observed a tendency to inhibit the expression of other antioxidant enzymes, suggesting potential effects on TXNRD1/2.
To date, no side effects have been observed with Exo derived from ASC.
The reason for selecting BCl-xL (anti-apoptotic factor) and Bax (pro-apoptotic factor) in qRT-PCR is that both Bcl-xL and Bax belong to the Bcl-2 protein family, which plays a crucial role in the apoptosis and survival pathways centered around the mitochondria. These proteins regulate the balance between cell survival and death and can also influence mitochondrial antioxidant mechanisms.
15) BTW, a high-quality graphical abstract was missing. Good to have one.
Answer> Thank you for your suggestion. We will prepare high-quality graphic abstract. However, because of the time limit of revision deadline. We have no choice to submit revision without a high-quality graphic abstract. We would like to resubmit with additional graphic abstract if we can have more time to prepare.

Round 3
Reviewer 3 Report
Comments and Suggestions for Authors
Good to have many improvements on the revised manuscript. Some remaining minor issues could be modified as follows,
1) The average body weight of experimental mouse in the study, should be clearly described in the main texts and if necessary also in the supporting materials. Authors should not hide this information in the text, but only disclosed it just in the supplementary part. No reasons. Is that information unnecessary or unimportant? The mean mice body weight used for the tests, matters a lot in reasoning and supporting your conclusions. The smaller mouse was obviously small, for the very early glance I took, immediately I thought, I guess other people may promptly notice this.
2) A Good graphical abstract is of course a key to your story. to get people, to read your paper. It is up to you. I am totally fine with it being published without a GA.
Author Response
1) The average body weight of experimental mouse in the study, should be clearly described in the main texts and if necessary also in the supporting materials. Authors should not hide this information in the text, but only disclosed it just in the supplementary part. No reasons. Is that information unnecessary or unimportant? The mean mice body weight used for the tests, matters a lot in reasoning and supporting your conclusions. The smaller mouse was obviously small, for the very early glance I took, immediately I thought, I guess other people may promptly notice this.
=> Thank you for your comment. we revised Figure 4. to display the average weight change after the initiation of treatment.
2) A Good graphical abstract is of course a key to your story. to get people, to read your paper. It is up to you. I am totally fine with it being published without a GA.
==> We uploaded GA. Thank you again.